# Malassezia Folliculitis following Triple Therapy for Cystic Fibrosis

**DOI:** 10.3390/medicina58091204

**Published:** 2022-09-02

**Authors:** Federica Li Pomi, Luca Di Bartolomeo, Mario Vaccaro, Maria Lentini, Simona Cristadoro, Maria Cristina Lucanto, Mariangela Lombardo, Stefano Costa, Francesco Borgia

**Affiliations:** 1Department of Clinical and Experimental Medicine, Section of Dermatology, University of Messina, 98122 Messina, Italy; 2Department of Human Pathology of Adult and Childhood Gaetano Barresi, University of Messina, 98122 Messina, Italy; 3Cystic Fibrosis Center, Gastroenterology and Cystic Fibrosis Unit, Department of Human Pathology of Adult and Childhood Gaetano Barresi, University of Messina, 98122 Messina, Italy

**Keywords:** cystic fibrosis, skin, adverse drug reaction, folliculitis, Malassezia, elexacaftor/tezacaftor/ivacaftor, microbiome

## Abstract

Triple-combination therapy with elexacaftor, tezacaftor and ivacaftor has been recently approved for cystic fibrosis patients with at least one F508*del* mutation in the transmembrane conductance regulator of the cystic fibrosis gene. Among the adverse events of elexacaftor, tezacaftor and ivacaftor, the cutaneous ones have been rarely reported, mainly dealing with urticarial-like rashes. On this topic, we report two cases of Malassezia folliculitis following triple therapy administration in two young females. In the first patient, a papulopustular rush appeared before the folliculitis while in the second patient it was not preceded by other skin manifestations. The diagnosis was confirmed both by dermoscopy and histology. The prompt response to systemic antimycotic drugs provided further evidence for the causative role of Malassezia, requiring no discontinuation of cystic fibrosis therapy. We could hypothesize that the triple regimen treatment may induce changes in the skin microbiome, potentially able to favor colonization and proliferation of Malassezia species. Physicians should be aware of such associations to allow prompt diagnosis and early interventions, avoiding useless drug removal.

## 1. Introduction

Cystic fibrosis (CF) is an autosomal recessive inherited disease, with the highest prevalence in the Caucasian population [1]. CF is a progressive, life-shortening disorder, characterized by defects in chloride ion transport which lead to mucosal hyperconcentration in the respiratory, digestive and reproductive systems and malabsorption of chloride and sodium in the sweat glands [2,3]. The disease usually presents in childhood with recurrent pulmonary infections and exocrine pancreatic failure causing gradual loss of pulmonary function and poor weight gain, respectively [2]. It is caused by mutations in the transmembrane conductance regulator of the cystic fibrosis regulator (CFTR) gene. Approximately 2000 mutations of the CFTR gene are known, the most common of which is the F508*del* mutation, estimated to be present in 90% of CF patients [1,2]. A triple-combination CFTR modulator regimen with ivacaftor (IVA), elexacaftor (ELX), and tezacaftor (TEZ) has been recently approved for CF patients with at least one F508*del* mutation [4]. While ELX and TEZ are small-molecule correctors that increase CFTR cell-surface expression, IVA improves the activity of the defective CFTR protein [4]. These molecules’ combination has shown dramatic improvement in lung function, sweat chloride concentration and patient-reported quality of life [3,4]. CFTR modulator therapy has shifted the CF treatment from a therapy relieving symptoms to a therapy that also restores the function of the CFTR protein, thus expanding the field of personalized medicine [5]. However, adverse events to ELX/TEZ/IVA have been reported in the literature. Among them, the cutaneous ones have been rarely observed, mainly dealing with urticarial-like rash. On this topic, we report two novel adverse reactions not previously described in patients treated with ELX/TEZ/IVA.

## 2. Case Reports

### 2.1. Case 1

A 19-year-old Caucasian woman affected by CF and homozygous for the F508*del* mutation was referred to our Dermatology Unit for the sudden onset of a widespread skin rash. Familial and personal history was negative for cutaneous diseases. She never suffered from acne and folliculitis. She had been previously treated with dual CFTR modulator therapies with Lumacaftor (LUM)/IVA for two years with gradual decline of efficacy. Therefore, she was started on ELX/TEZ/IVA treatment with the following schedule: two tablets containing 100 mg/50 mg/75 mg per day respectively in the morning and one IVA tablet of 150 mg per day in the evening, according to European Medicines Agency guidelines. On day 3 of triple therapy, she developed a widespread itchy rash localized to the abdomen and upper limbs progressively involving the buttocks and lower limbs. She was not undergoing any other treatment and she has had no known infections in the past month (Figure 1a). No bullae were noted and Nikolsky sign was negative. Mucosal surfaces were spared. She did not complain of other cutaneous or systemic symptoms. ELX/TEZ/IVA therapy was stopped and oral antihistamine and topical steroid ointment were administered. The rash resolved within 15 days. ELX/TEZ/IVA was therefore cautiously re-administered. Ten days after, the patient developed a papulo-pustular rash initially localized on the upper limbs, gradually involving trunk and face. Physical examination revealed a plethora of erythematous, pinhead-sized papular lesions and follicular-centered pustules. No blackheads were detected on physical examination. Concomitantly, hypopigmented macules and patches of pityriasis versicolor appeared on the patient’s trunk (Figure 1b). Dermoscopic evaluation revealed the presence of folliculocentric pustules with surrounding erythema (Figure 1c). A 3.5 mm punch biopsy performed on a pustule on the right shoulder revealed spherical to oval yeast-like organisms at Periodic acid-Schiff (PAS) stain, which confirmed the diagnosis of Malassezia folliculitis. CF treatment was therefore continued. The patient was treated with oral fluconazole 100 mg/day for 14 consecutive days with improvement at 1 month’s follow-up.

### 2.2. Case 2

A 24-year-old woman with CF was initiated on ELX/TEZ/IVA (100 mg/50 mg/75 mg) treatment: two tablets per day in the morning and one tablet of IVA 150 mg in the evening. She had been previously treated with CFTR modulator therapy with IVA for five years with gradual loss of response. Nine days after starting triple therapy, an itchy papulopustular rash mainly located on the back was observed (Figure 2a). Follicular pustules with surrounding erythema were detectable at dermoscopic examination (Figure 2b). Familial and personal history was negative for cutaneous diseases. She never suffered from acne and folliculitis. Punch biopsy was performed on a single pustule. Histological examination with hematoxylin and eosin (H–E) stain revealed features of folliculitis with dilatation of the infundibulum and plugging of the follicular ostium with keratin and cellular debris. In the perinfundibular dermis, granulomatous reaction with isolated multinucleated giant cells was detected (Figure 2c). Silver methenamine stain highlighted round yeast-like organisms at the top of the follicle and hyphae in the stratum corneum of the follicular infundibula, thus confirming the presence of Malassezia in the follicular pustule (Figure 2d). Discontinuation of the drug was deemed unnecessary. The patient was treated with oral bilastine 20 mg/day and fluconazole 100 mg/day for 14 consecutive days with progressive improvement of the folliculitis.

## 3. Discussion

ELX/TEZ/IVA is approved for patients with CF who have at least one F508 del mutation in the CFTR gene. Its efficacy has been demonstrated in four studies involving patients aged 6 years and above by measuring the increase in the percent predicted forced expiratory volume in one second, known as ppFEV1, an outcome parameter of lung disease progression [4,6,7,8]. Phase 3 clinical trials reported “rash” in 4% up to 11% of participants, although details on clinical characteristics and outcomes have not been fully described [4,6,7,8]. Real-life experience is limited to a few cases of widespread urticarial-like rash (eight cases) [9,10,11,12,13,14,15,16], morbilliform rash (one case) [9], acneiform reaction (one case) [17], and maculopapular rash (two cases) [18] with onset between 1 and 120 days after the start of the treatment. In all patients, a triple regimen was stopped with progressive resolution of the rash. In 10 patients, the drugs were re-administered, with new development of rash in six cases. Slow oral desensitization was effective to prevent rash recurrence in five of them, while in one patient triple therapy was continued with prescription of topical antibiotics [17] (Table 1). No clear recommendation emerges from current literature about the management of ELX/TEZ/IVA-related rashes. Rashes appear to be more severe at the onset, while recurrences seem to be milder, usually requiring no discontinuation of the therapy. Systemic steroids and antihistamines may be helpful to control the symptoms in severe cases. Given the therapeutic benefit of ELX/TEZ/IVA and its increasing use, oral desensitization may be considered as an option to avoid full discontinuation of the drugs, thus allowing patients to receive long-term treatment.

To the best of our knowledge, cases of Malassezia folliculitis related to ELX/TEZ/IVA have not yet been reported in the literature. We described two cases with different onset modalities. In the first one, a follicular papulopustular rash occurred before Malassezia folliculitis, which developed after ten days from re-administration of triple therapy. In the second one, a Malassezia folliculitis rash was not preceded by other manifestations arising nine days after beginning triple therapy. However, both patients benefited from antimycotic therapy. Malassezia folliculitis is an underdiagnosed follicular papulo-pustular eruption caused by Malassezia yeast overgrowth, which is commonly present in the infundibulum of the sebaceous glands as commensals of normal skin flora. Alterations in sebaceous gland activity and follicular occlusion may be predisposing factors to Malassezia folliculitis, as well as the imbalance of normal skin flora caused by broad-spectrum antibiotics [19,20]. Similarly, predisposing factors include immunosuppression, oral corticosteroids or broad-spectrum antibiotics [19]. Malassezia folliculitis is often misdiagnosed as acne or bacterial folliculitis, even if comedones are absent and itching is a common symptom. Diagnosis is based on the clinical picture, histopathological findings and response to antimycotic therapy [21]. H–E staining reveals features of folliculitis, while keratinous material, cellular debris and inflammatory infiltrate of mononuclear cells are usually observed in the perifollicular dermis [21]. In suspected Malassezia folliculitis cases, PAS staining should be considered when yeast cells are not detected by H-E staining [22]. Response to antifungal therapy can be another useful diagnostic tool. Oral antifungals have been shown to be more effective in therapeutic response, while topical antifungals are important for maintenance and prophylaxis of relapses [23]. ELX/TEZ/IVA therapy, acting on the chlorine and sodium channels, can cause microbiome and metabolome modification in the respiratory tract mucus [11]. Triple therapy also acts on the sweat glands’ duct, promoting the reabsorption of sodium and chlorine. Accordingly, we could hypothesize that ELX/TEZ/IVA may induce changes in the skin microbiome, potentially able to favor colonization and proliferation of Malassezia species. Currently, no studies about cutaneous hydration, pH and sebometry of FC patients are reported in the literature. It would be useful to measure these parameters before and after triple therapy in order to register any variations in the skin microbiota. Moreover, investigating the effect of ELX/TEZ/IVA on skin parameters and microbial flora, in a larger case series, could provide better insight into ELX/TEZ/IVA-induced skin rashes.

## 4. Conclusions

We reported two cases of Malassezia folliculitis following ELX/TEZ/IVA therapy. We suggest that a triple therapy regimen may induce changes in the skin microbiome, which could potentially favor colonization and proliferation of Malassezia species, commensals of normal skin flora. Nevertheless, further studies are needed to clarify the effects of ELX/TEZ/IVA treatment on the skin in order to improve physicians’ diagnostic capability in cases of drug-related rashes. Moreover, a better understanding of these effects could improve the management of patients, thus avoiding useless drug removal, with consequent increased morbidity and mortality.

## Figures and Tables

**Figure 1 medicina-58-01204-f001:**
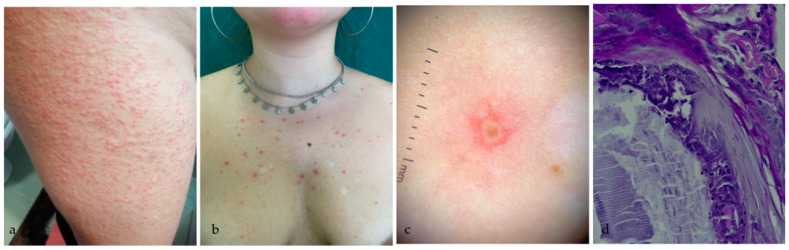
(**a**) First rash: plethora of erythematous pinhead-sized papular lesions, many of which were centered by follicular pustules; (**b**) second rash: papulopustular lesions involving décolleté with hypopigmented macules and patches of pityriasis Versicolor; (**c**) Dermoscopy (10×) of a folliculocentric pustule with surrounding erythema; (**d**) PAS (40×) re-vealed spherical to oval yeast-like organisms.

**Figure 2 medicina-58-01204-f002:**
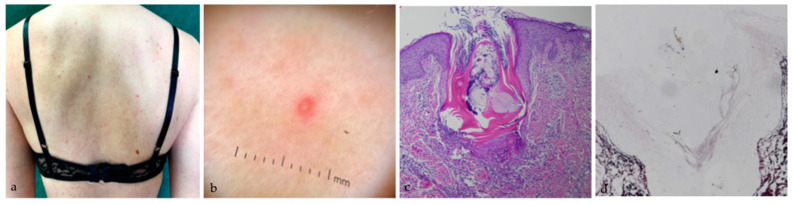
(**a**) Malassezia folliculitis involving the back; (**b**) dermoscopy (10×) shows follicular pustule with surrounding erythema; (**c**) H-E stain (20×) reveals features of folliculitis with dilatation of the infundibulum and plugging of the follicular ostium with keratin and cellular debris. (**d**) Silver methenamine stain (20×) highlights a ruptured follicle containing numerous Malassezia yeasts.

**Table 1 medicina-58-01204-t001:** Previous studies reporting cutaneous adverse reactions to Elexacaftor/Tezacaftor/Ivafactor.

First Author, Reference, Year	Sex	Age	Rash-Type	Onset of the Rash (Days)	Re-Administration	New Rash	Therapeutic Decision
Leonhardt K [9], 2021	F F	20 47	Morbilliform Urticarial-like	7 9	Yes Yes	Yes, after 11 days No	Desensitization Interruption
Goldberg RH [12], 2021	M	12	Urticarial-like	4	No		Interruption
Cheng A [13], 2022	M	?	Urticarial-like	9	Yes	Yes, after 24 h	Desensitization
Stashower J [14], 2021	F	24	Urticarial-like	7	No		Discontinuation
Hu MK [10], 2020	F	24	Urticarial-like	8	Yes	No	Continuation
Balijepally R [15], 2022	F F	48 37	Urticarial-like Urticarial-like	7 6	Yes Yes	Yes, only on full-dose Yes, after 8 h	Desensitization Desensitization
Loyd I [16], 2022	M	7	Urticarial-like	7	Yes	Yes, the same day	Desensitization
Breneman A [17], 2021	M	29	Acneiform	120	Yes	Yes, after 9 days	Continuation
Diseroad E [18], 2022	F M	12 14	Maculo-papular Maculo-papular	1 8	Yes Yes	No No	Desensitization Desensitization

## Data Availability

Not applicable.

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
