# Peer review of "Malassezia Folliculitis following Triple Therapy for Cystic Fibrosis"

_medicina, 2022, doi:10.3390/medicina58091204_

Round 1
Reviewer 1 Report
It is very dificult to prove that an agent like P. ovale that is a part of normal follicular microbiota, is the ethiological agent of the folliculitis. Sometimes the response to the antimycotic treatment is considered as a prove of P. ovale role in the folliculitis, more than the finding of the fungus in the follicle. Please add this aspect in the discussion. In fig. 1 and 2, provide a picture of HE stained section demostrating the presence of folliculitis.
Reviewer 2 Report
This study is very interesting with plenty of excellent results. It has a lot of lab work. According to my opinion, it merits publication in Medicina.
However, the below-mentioned points need to be addressed before taking it to the further stage.
1. English editing is suggested. Punctuation related errors are there, without any punctuation authors have started writing in Capital letters, this kind of silly mistakes needs to be corrected.
2. The title is long, a title should be brief and concise
3. Avoid to use abbreviation in the abstract
4. The abstract is one of the most important parts of the manuscript. Well written summaries improve the impact of the paper and expedite Peer Review. Abstract should be specific to the findings, aims and conclusions. Use short punchy statements with:
- Comparative data should be inserted
- Concise conclusions in one or two sentences.
5. Key-words: avoid to repeat the same words contained in the text
6. Without a thorough literature review, referees and editors are much less likely to accept that the research is sufficiently topical or original. Furthermore, by citing recent articles from currently active researchers from internationally available journals your research is much more likely to attract attention and to be read and cited.
You should aim to improve the focus of both the Introduction and Discussion sections upon the latest research. In what is a very active area of research too many of the articles cited are more than 5 years old and too few less than 3 years old (2020-2022). Greater emphasis should be made of the most recent research from authoritative international medical journals.
7. Authors used only 3 references, Please use at least (10-12) references
8. L32-33 please develop this sentence with proper reference’s
9. L43-45rewrite this sentence
10. authors should stress the aim of the study; please rewrite it
11. L 56, (ELX/TEZ/IVA treatment (100 mg/50 mg/75 mg per day) please explain the used concentration
12. Why authors used this concentration in this case
13. Please improve the fig 2 in term of quality
14. What is the link between studied cases
15. Authors should compare the results with other recent works
16. The optimal Conclusion should include:
• A summary of your key findings.
• A highlight of your hypothesis, new concepts, and innovations.
• A summary of key improvements compared to findings in the literature [provide a couple of references to indicate key improvements].
• Your vision for future work.
17. Author must be following the publication guideline of the journal before the final submission for publication of your article.
18. The abbreviations used should be written with full form when used first time.
